# Gene Expression and Epigenetic Changes in Mice Following Inhalation of Copper(II) Oxide Nanoparticles

**DOI:** 10.3390/nano10030550

**Published:** 2020-03-18

**Authors:** Pavel Rossner, Kristyna Vrbova, Andrea Rossnerova, Tana Zavodna, Alena Milcova, Jiri Klema, Zbynek Vecera, Pavel Mikuska, Pavel Coufalik, Lukas Capka, Kamil Krumal, Bohumil Docekal, Vladimir Holan, Miroslav Machala, Jan Topinka

**Affiliations:** 1Department of Nanotoxicology and Molecular Epidemiology, Institute of Experimental Medicine of the Czech Academy of Sciences, 14220 Prague, Czech Republic; kristyna.vrbova@iem.cas.cz (K.V.); vladimir.holan@iem.cas.cz (V.H.); 2Department of Genetic Toxicology and Epigenetics, Institute of Experimental Medicine of the Czech Academy of Sciences, 14220 Prague, Czech Republic; andrea.rossnerova@iem.cas.cz (A.R.); tana.brzicova@iem.cas.cz (T.Z.); alena.milcova@iem.cas.cz (A.M.); jan.topinka@iem.cas.cz (J.T.); 3Department of Computer Science, Czech Technical University in Prague, 12135 Prague, Czech Republic; klema@fel.cvut.cz; 4Department of Environmental Analytical Chemistry, Institute of Analytical Chemistry of the Czech Academy of Sciences, 60200 Brno, Czech Republic; vecera@iach.cz (Z.V.); mikuska@iach.cz (P.M.); coufalik@iach.cz (P.C.); capka@iach.cz (L.C.); krumal@iach.cz (K.K.); docekal@iach.cz (B.D.); 5Department of Chemistry and Toxicology, Veterinary Research Institute, 62100 Brno, Czech Republic; machala@vri.cz

**Keywords:** copper(II) oxide nanoparticles, inhalation, mouse, gene expression, DNA methylation

## Abstract

We investigated the transcriptomic response and epigenetic changes in the lungs of mice exposed to inhalation of copper(II) oxide nanoparticles (CuO NPs) (8 × 10^5^ NPs/m^3^) for periods of 3 days, 2 weeks, 6 weeks, and 3 months. A whole genome transcriptome and miRNA analysis was performed using next generation sequencing. Global DNA methylation was assessed by ELISA. The inhalation resulted in the deregulation of mRNA transcripts: we detected 170, 590, 534, and 1551 differentially expressed transcripts after 3 days, 2 weeks, 6 weeks, and 3 months of inhalation, respectively. Biological processes and pathways affected by inhalation, differed between 3 days exposure (collagen formation) and longer treatments (immune response). Periods of two weeks exposure further induced apoptotic processes, 6 weeks of inhalation affected the cell cycle, and 3 months of treatment impacted the processes related to cell adhesion. The expression of miRNA was not affected by 3 days of inhalation. Prolonged exposure periods modified miRNA levels, although the numbers were relatively low (17, 18, and 38 miRNAs, for periods of 2 weeks, 6 weeks, and 3 months, respectively). Kyoto Encyclopedia of Genes and Genomes (KEGG) pathways analysis based on miRNA–mRNA interactions, revealed the deregulation of processes implicated in the immune response and carcinogenesis. Global DNA methylation was not significantly affected in any of the exposure periods. In summary, the inhalation of CuO NPs impacted on both mRNA and miRNA expression. A significant transcriptomic response was already observed after 3 days of exposure. The affected biological processes and pathways indicated the negative impacts on the immune system and potential role in carcinogenesis.

## 1. Introduction

Copper(II) oxide nanoparticles (CuO NPs) may enter the body by means of inhalation, dermal or ingestion. In human exposure, the inhalation route is the most significant one, particularly for occupational exposures. In the body, nanoparticles (NPs) may reach the lungs where they are deposited, and additionally other organs including the central nervous system. Their presence usually causes oxidative stress or inflammatory responses.

Copper (Cu) is a biogenic element with many biological functions that include the appropriate growth, development, and maintenance of bones, connective tissues and various organs [1]. It is further involved in the stimulation of the immune system in the protection against infection and in the repair of injuries and healing. It plays an important role as a cofactor of some enzymes (e.g., superoxide dismutase and cytochrome oxidase) and is involved in cell signaling and cell proliferation. In addition, its metabolism is critical for tumor progression [2]. Nanosized Cu possesses antimicrobial and antiviral properties and is therefore used in face masks, wound dressings, and socks [3]. It is applied in conductive films, lubrication, nanofluids, gas sensors, solar cells, lithium batteries, and catalysts. Cu-based nanostructures are also used in various electronic devices, e.g., inkjet printers [4]. In comparison with other metal-based NPs (e.g., Ag or Zn), Cu-based NPs are less commonly used: the global production of CuO NPs was 570 tons in 2014 and it is estimated to increase to 1600 tons in 2025 [5]. The toxicity of Cu-based NPs depends on the solubility of the material that is affected by pH, dissolved organic carbon content, and water hardness [3].

Although the production of CuO NPs is at present relatively low, their properties and behavior in biological systems make them an important subject for toxicological investigation. The theoretical mechanisms of CuO NPs toxicity include: (1) release of Cu ions from the surface of NPs, (2) oxidative damage mediated by Fenton-type reactions [reactive oxygen species (ROS) generation] induced by particles, and (3) cell death associated with autophagy caused by CuO NP exposure. While metal oxide NPs (e.g., Cu, Zn, Ti, Fe, and Si) share similarities in their biological effects often mediated by ROS production, CuO NPs have been shown to have the highest potential to induce cytotoxicity and DNA damage in vitro [6]. This seems to be related to the ability of CuO NPs to better overwhelm the antioxidant defenses of the organism.

The biological effects of the various methods of CuO NPs pulmonary delivery, including intratracheal instillation, oropharyngeal instillation, or whole-body inhalation, were investigated in both rats and mice (reviewed in [1,6]). The majority of the studies reported inflammatory responses in the lungs that included neutrophil and eosinophil infiltration after acute exposures, and neutrophilic, neutrophilic/lymphocytic, eosinophilic/fibrotic/granulomatous, and fibrotic granulomatous inflammation after chronic exposures. Pulmonary administration may even cause systemic effects affecting the liver, kidney, or spleen, as shown by Liu et al. [7] and in our own recent study [8]. Inflammatory processes may induce DNA damage which may cause mutations and tumor formation in the lungs, as demonstrated by Yokohira et al. in a study in rats [9].

Despite the potential serious health effects of human exposure to CuO NPs, only one study that focused on the analysis of whole genome transcriptional changes, following the inhalation of CuO NPs, has been published [10]. Studies of global gene expression changes are valuable as they have the potential to reveal a wide range of processes which are affected by a respective exposure, unable to be detected by analyses focused on a defined, limited number of pathological changes. To address these gaps, we conducted a study in which we exposed ICR mice to CuO NPs by continuous whole-body inhalation. We used four time periods to mimic acute and chronic exposure and analyzed global mRNA expression changes and epigenetic alterations by analysis of miRNA and global DNA methylation changes. Moreover, we identified mRNA targets of miRNAs found to be deregulated in our study and performed an analysis of pathways affected by CuO NPs exposure.

## 2. Materials and Methods

### 2.1. CuO NPs Generation, Exposure of Mice, Analysis of Cu and CuO NPs in the Lungs

The generation of CuO NPs, mice exposure and analysis of Cu and CuO NPs in the lungs, were reported in our previous study [8]. Briefly, CuO NPs were continuously generated by using the thermal decomposition of metal organic precursor copper (II) acetylacetonate and a subsequent oxidation at a temperature of 700 °C. CuO NPs were diluted with a U-HEPA filtrated air stream, and then split into two streams at a ratio of 1:1. Prior to entering the inhalation chamber, both streams were further diluted with a stream of purified, humidified air and used for whole-body inhalation experiments. Female ICR mice (8 animals/group) were exposed in a whole-body inhalation chamber [11] to 8 × 10^5^ CuO NP/cm^3^ (geometric mean diameter 29.4 nm) for 3 days, 2 weeks, 6 weeks, and 3 months; the control animals inhaled clean air. The conditions of inhalation were selected based on our previous study [8]. At the selected time intervals mice were sacrificed by chloroform inhalation and the lungs were collected for further analyses. The animal experiments were approved by the Ethical Board of the Institute of Analytical Chemistry in Brno.

An analysis of Cu content was performed using electrothermal atomic absorption spectrometry, employing AAnalyst 600 (Perkin-Elmer Inc., Wellesley, MA, USA) instrumentation. The presence of CuO NPs in the lungs was assessed using scanning transmission electron microscopy (STEM) and scanning electron microscopy-energy-dispersive X-ray spectroscopy (SEM-EDX). The sections were examined under a Philips EM 208 S Morgagni transmission electron microscope (FEI, Brno, Czech Republic). For the SEM-EDX analysis, a Magellan 400 SEM (FEI) equipped with an Energy Dispersive Spectroscopy Detector Octane Elect Super (EDAX, Baltimore, MD, USA) was used.

The number concentration and the size distribution of CuO NPs in the inhalation chambers were measured continuously in the size range of 7.64–229.6 nm using a Scanning Mobility Particle Sizer Spectrometer (SMPS; TSI, Shoreview, MN, USA) at 5 min intervals.

### 2.2. DNA, RNA, and miRNA Extraction

Lung tissue was disrupted using a mortar and pestle, and homogenized with a needle and syringe. From tissue homogenates, the total RNA and DNA was extracted using AllPrep DNA/RNA/miRNA Universal Kit (QIAGEN Manchester Ltd; Manchester, UK). The extraction protocol followed the manufacturer’s instructions. The concentration of nucleic acids was determined by Nanodrop ND-1000 Spectrophotometer (Thermo Fisher Scientific; Waltham, MA, USA); RNA integrity number (RIN) was assessed by Agilent Bioanalyzer (Agilent Technologies Inc.; Santa Clara, CA, USA) with RNA 6000 Nano kit (Agilent Technologies Inc.; Santa Clara, CA, USA). RIN values were > 7.5 in all samples.

### 2.3. Whole Genome Transcriptome Analysis by Next Generation Sequencing

RNA samples (concentration > 260 ng/µl, RNA integrity number > 7.5) were used for RNA libraries preparation using (Poly(A)RNA Selection, Lexogen Sense Total RNA-Seq Library Prep Kit (Lexogen GmbH; Vienna, Austria). For sequencing, NextSeq® 500/550 High Output Kit v2 (75 cycle) and NextSeq 500/550 system (Illumina, Inc.; San Diego, CA, USA) were used. The reactions were performed according to the manufacturers’ recommendations.

### 2.4. miRNA Expression Analysis

Small RNA libraries were prepared from 150 ng of total RNA using Qiaseq miRNA library kit (QIAGEN Manchester Ltd; Manchester, UK). This kit allows detection of not only miRNAs, but also other small RNAs, including piRNAs. In our data, a vast majority of results consisted of deregulated miRNAs, with a small proportion of piRNAs. Thus, “miRNA expression” reported here denotes expression of both miRNAs and piRNAs. Briefly, adaptors were ligated to the 3′ and 5′ ends of all miRNAs. After adaptor ligation, miRNA was reverse transcribed to cDNA and the library was amplified (95 °C 15 min, (95 °C 15 s, 60 °C 30 s, 72 °C 15 s, 16×), 72 °C 2 min, 4 °C for at least 5 min) on a Mastercycler^®^ Nexus (Eppendorf; Hamburg, Germany). Small RNA libraries were validated on a Fragment Analyzer (Agilent Technologies Inc.; Santa Clara, CA, USA) with a High sensitivity NGS kit. Sequencing of sRNA libraries was performed on MiSeq system (Illumina, Inc.; San Diego, CA, USA) using MiSeq Reagent Kit V3 (Illumina, Inc.; San Diego, CA, USA).

### 2.5. Global DNA Methylation Analysis

The content of 5-methylcytosine (5-mC) was assessed in DNA samples (100 ng/well) using 5-mC DNA ELISA Kit (Zymo Research; Irvine, CA, USA) following the manufacturer’s instructions. The samples were analyzed in triplicate. The results were expressed as % 5-mC/total cytosine content.

### 2.6. Data Analysis

An NGI-RNAseq pipeline (https://github.com/SciLifeLab/NGI-RNAseq) was used for RNA sequencing data. It pre-processed raw data from FastQ inputs (FastQC, Trim Galore!), aligned the reads (HiSAT2) using reference genome Mus musculus assembly GRCm38.91, Ensembl annotation version 91, generated gene counts (featureCounts) and performed extensive quality-control on the results (RSeQC, dupRadar, Preseq, edgeR, and MultiQC). DESeq2 with default parameter settings was applied to normalize read counts and to identify differences in gene expression between sample groups. The standard deviations for the numbers of differentially expressed mRNAs were estimated with bootstrapping (25 times repeated sampling with replacement carried out independently in each of the treatment groups). A functional enrichment analysis of differential gene expression data was performed by ToppFun tool, using the features “Biological process” and “Pathway” [12].

Differential miRNA expression count data were generated by QIAseq miRNA Quantification, and the UMI counts were taken to compensate for sequencing bias. The DESeq2-package was used to test for differential expression by the application of negative binomial generalized linear models. A *p* < 0.02 was used as the cut-off for statistically significant deregulated miRNAs between sample groups.

mRNA-miRNA correlation analysis was performed with the R package miRComb [13]. First, the sets of differentially expressed mRNAs and miRNAs were identified (the detection thresholds were adjusted pval < 0.1, FC > 1.5; the detection method was DESeq). Correlations between all the mRNA and miRNA candidate pairs were then calculated. We used the default parameter settings proposed in miRComb (Pearson method). Only the pairs with a negative correlation were considered. The pairs with a significantly adjusted correlation *p*-value were reported and used to construct the mRNA-miRNA interaction networks. We extended the interaction networks using information regarding validated and predicted miRNA targets. These targets were taken from the multiMiR package [14] that comprises of 14 different human and mouse miRNA target databases. The predicted and validated targets were used to highlight the edges in mRNA-miRNA interaction correlation networks and also helped to identify miRNAs with the highest number of deregulated targets. The R package edgeR served to report the enriched Kyoto Encyclopedia of Genes and Genomes (KEGG) pathways [15].

## 3. Results

### 3.1. mRNA Expression Changes Induced by CuO NPs Inhalation

Exposure to CuO NPs was associated with the deregulation of mRNA expression. The numbers of differentially expressed genes increased with the length of exposure, although the change was not consistent and standard deviations for these numbers are large as a consequence of small sample sizes (170 ± 84, 590 ± 331, 534 ± 172, and 1551 ± 387 deregulated genes for 3 days, 2 weeks, 6 weeks, and 3 months of exposure, respectively) (Appendix A). The number of upregulated genes was generally higher than those that were downregulated; this difference ranged from 1.37-fold to 3.27-fold, depending on the exposure period. The distribution of common and unique deregulated genes is shown in the Venn diagram (Figure 1). Only two transcripts were common for all of the exposure periods (Hmox1, Slc7a11). In contrast, 159 genes were commonly deregulated for 2 weeks, 6 weeks, and 3 months exposure. Interestingly, few common transcripts were found for any comparison in which 3 days exposure was included. This suggests that the effects associated with the shortest (acute) exposure, substantially differed from those induced by longer exposure periods. The list of the common and unique deregulated genes is reported in Appendix A.

In agreement with findings for differentially expressed mRNAs, the functional enrichment analysis revealed differences between biological processes and pathways deregulated after 3 days exposure and longer treatments. The short inhalation period mostly affected collagen metabolism and extracellular matrix organization. Prolonged exposure caused the deregulation of immune response-related processes including cytokine production and cell activation (selected biological processes and pathways are reported in Table 1). We further investigated the biological processes and pathways unique for individual, longer exposure periods. For 2 weeks of treatment, the processes associated with immune response and cell death were deregulated, 6 weeks of inhalation affected cell division and cell cycle regulation and 3 months of treatment involved changes in cell adhesion, migration, and cytokine production (selected unique biological processes are shown in Table 2).

### 3.2. Changes of miRNA Expression Affected by CuO NPs

The short, 3-day inhalation period had no effect on miRNA expression, but the longer exposure periods were associated with significant changes in miRNA levels. We observed 17, 18, and 38 deregulated miRNAs for 2 weeks, 6 weeks, and 3 months exposure, respectively (Appendix A). Similarly to mRNA expression, the number of upregulated miRNAs was higher than those that were downregulated. The greatest difference (4.6-fold) was observed for the 2-week inhalation period, followed by 6 weeks (1.57-fold) and 3 months (1.11-fold) exposure. The numbers of common and unique deregulated miRNAs are shown in Figure 2. There were nine miRNAs common for all three exposure periods. The number of unique deregulated miRNAs differed by the exposure time: the highest proportion of such miRNAs (61% of all deregulated miRNAs) was detected after 3 months of inhalation. The common and unique deregulated miRNAs are shown in Appendix A.

The interactions between experimentally validated, significantly deregulated miRNAs and mRNAs are reported in Appendix A. The number of significant results increased again with the length of exposure: 148 mRNAs targeted by three miRNAs, 575 mRNAs targeted by 18 miRNAs and 1542 mRNAs targeted by 15 miRNAs were found after 2 weeks, 6 weeks, and 3 months of inhalation, respectively. A summary of miRNAs involved in the interactions and the respective number of target mRNAs are reported in Table 3. The highest number of miRNAs was affected by mmu-miR-449a-5p (2 weeks of inhalation), mmu-miR-448-3p (6 weeks exposure) and mmu-miR-1264-3p, mmu-miR-1298-5p, mmu-miR-144-5p (6 weeks and 3 months of treatment). We further studied miRNA–mRNA interactions using interaction networks. These networks graphically depict targets for individual miRNAs and the potential relationship with targeted mRNAs. The density of the networks and number of visualized interactions depend on the selected, adjusted *p*-value for interactions and a score and generally increase with a higher *p*-value and lower score. The score is computed as a measure of impact of miRNA on respective mRNA expression and a higher score means a tighter miRNA–mRNA association [13]. The results for individual miRNAs and their targets for adjusted *p*-value <0.001 and score ≥ 3 are visualized in Figure 3. Using these conditions, the highest density of interactions was found with 6 weeks of inhalation, this was as expected considering the highest number of miRNAs significantly targeting mRNAs was for this exposure period. In these interactions, five miRNAs were involved (mmu-miR-135b-3p, mmu-miR-135b-5p, mmu-miR-448-3p, mmu-miR-1264-3p, mmu-miR-1298-5p). Interestingly, despite the highest number of targeted mRNAs after 3 months of inhalation, the density of interaction networks was low and only one miRNA was detected (mmu-miR-1264-3p). For 2 weeks of inhalation, mmu-miR-135b-5p and mmu-miR-449a-5p participated in miRNA–mRNA interactions. The details on miRNA–mRNA interactions depicted in Figure 3 are provided in Appendix A.

The KEGG pathway analysis based on miRNA–mRNA interactions revealed 3, 25, 1, and 20 significantly deregulated pathways after 3 days, 2 weeks, 6 weeks, and 3 months of inhalation, respectively (Table 4). Most of these pathways were unique for individual exposure periods. Potentially important results include axon guidance, gap junction, dilated cardiomyopathy, Wnt signaling, calcium signaling (2 weeks), and immune response related pathways (3 months). In addition, olfactory transduction was induced by 2 weeks and 3 months of treatment. The short exposure periods (3 days and 2 weeks) commonly affected the following pathways: ECM receptor interaction, focal adhesion and pathways in cancer. None of the pathways were commonly deregulated in all of the time periods.

### 3.3. Global DNA Methylation after CuO NPs Inhalation

The inhalation of CuO NPs had no significant effect on global DNA methylation in any of the exposure periods. Although a higher proportion of 5-methylcytosine/total cytosine in the exposed than in the controls could be seen after 3 days, 6 weeks, and 3 months of inhalation, the differences did not reach statistical significance. Furthermore, no significant trends in global DNA methylation changes relating to the exposure time were observed (Figure 4).

## 4. Discussion

In this study, we aimed to investigate the effects of whole-body CuO NPs inhalation on the expression of mRNA and epigenetic markers (expression of miRNA and global DNA methylation) in mice. The strategy of our experiment allowed us to assess the impact of both short-term acute inhalation and long-term chronic exposure.

The toxicity of CuO NPs is relatively high when compared with other metal oxide NPs. The negative effects are mostly mediated by ROS production followed by oxidative stress, destruction of mitochondrial membranes and the induction of downstream processes, resulting in autophagy and/or apoptosis [16], including e.g., activation of p53 which increases Bax/Bcl2 ratio. CuO NPs also cause cell cycle arrest and DNA damage manifested by ***γ***-H2AX formation. They further induce interleukin (IL)-8 production, which consequently causes the activation of NF-*κ*B pathway [17]. CuO NPs have been shown to bind to proteins via hydrogen bonds [18] causing changes of protein phosphorylation and ubiquitination, which plays an important role in cell signaling. They upregulate the expression of DNA repair proteins and directly interact with the structural elements of the cell (e.g., cytoskeleton). This may disrupt the mitotic spindle and cause aberrant cell division [16]. CuO NPs further directly interact with DNA, causing a replication arrest possibly leading to genome instability. Finally, copper has been shown to affect gene expression through epigenetic mechanisms, specifically by histone modification [19].

Despite many biological effects associated with CuO NPs exposure, studies on gene expression and epigenetic changes induced by these NPs are very rare. To date, only one study that focused on the analysis of whole genome transcriptional changes following the inhalation of CuO NPs [10] has been published. The authors exposed rats for 6 hours to two doses (3.3 and 13.2 mg/m^3^) of CuO NPs and analyzed the transcriptomics response 1 day post-exposure and after a 22 day recovery period. The exposure resulted in the deregulation of approximately 1000 genes in the high-dose group and approximately 200 genes in the low-dose group. After the recovery period, the number of deregulated genes dropped to around 20. The main processes affected by the exposure included cell proliferation/survival and inflammation. Interestingly, no oxidative stress-related pathways were affected. The authors observed proliferation of alveolar epithelial cells and detected the upregulation of CCl2 (monocyte chemoattractant protein 1), both on the level of mRNA and protein and the upregulation of Epithelial Cell Transforming 2 ECT2 oncoprotein. They concluded that short-term exposure to CuO NPs affected the processes associated with acute inflammation in the lungs and tumorigenesis. They pointed out that long-term exposure studies would be of interest.

Recently, we have shown that the inhalation of CuO NPs, affects the systemic immune response of the exposed mice [8]. Changes to the proportion of eosinophils, neutrophils, macrophages, and antigen-presenting cells (innate immunity), along with the modulation of proliferative and secretory activity of *T* cells were observed. The effects were dependent on the time of exposure: the production of cytokines increased after 3 days of inhalation, decreased after 2 weeks and then returned to the control levels at later time intervals. The proportion of cells of innate immunity decreased on day 3, increased after 2 weeks, and again decreased after 3 months of exposure.

In this study, we extended our investigation to gene expression and the epigenetic effects in the lungs of CuO NPs-exposed mice. In contrast to the study by Costa et al. [10], we analyzed whole genome mRNA expression changes, using next generation sequencing of the transcriptome, which allowed us to detect a higher number of deregulated mRNAs with greater sensitivity. We further assessed miRNA expression and focused on the detection of miRNA–mRNA interactions, using experimentally validated deregulated molecules. Although the number of differentially expressed mRNAs generally increased with the exposure time, the data suggested substantial differences between the effects induced by 3 days of inhalation and longer exposure periods. There was very little overlap between mRNAs commonly affected by all of the inhalation periods: only two genes (heme oxygenase 1, involved in oxidative stress response, and solute carrier family 7, encoding cystine/glutamate transporter) were deregulated, while for 2 weeks, 6 weeks, and 3 months exposures, 159 common transcripts were detected. Among these transcripts, immune response-related genes were prevalent [e.g., interleukin 1 beta, tumor necrosis factor, CD14, CD33, and CD80 antigens, chemokine (C–C motif) ligands, or toll-like receptor]. For the short exposure period, the greatest proportion of unique mRNA transcripts was found (138 unique of 170 total deregulated transcripts, 81%), followed by 3 months of exposure (1108 unique of 1551 total transcripts, 71%), while for 2 weeks and 6 weeks of treatments, the corresponding percentage was 38% and 35%, respectively. Genes deregulated after 3 days exposure included e.g., several collagen-encoding genes, zinc finger proteins, or chondroitin polymerizing factor. Two-week exposure resulted in changes of expression of apoptosis-relates genes (BCL2-associated athanogene 3, cell death inducing Trp53 target 1, and programmed cell death 1), as well as cytochrome P450 1B1 and superoxide dismutase 2 genes. After 6 weeks of inhalation, expression of genes involved in cell cycle was detected (cyclin-dependent kinase 1, aurora kinase *B*, spindle and kinetochore associated complex subunit 3, regulator of cell cycle, and cyclin A2). Three months of exposure was associated with deregulation of e.g., WNT1 inducible signaling pathway protein 2, jun B proto-oncogene, *B* cell leukemia/lymphoma 2 related protein A1a, mitogen-activated protein kinase 4, and B cell leukemia/lymphoma 3 that are all potentially implicated in carcinogenesis.

To further investigate the biological impacts of CuO NPs inhalation, we assessed the biological processes and pathways deregulated as a result of exposure. Again, a distinct response was found for the short inhalation period. It was generally associated with collagen metabolism and extracellular matrix organization. Collagen, the major part of extracellular matrix, is fundamental to maintain the structure and function of the lungs. In a recent study, collagen accumulation, along with the expression of the progressive fibrosis marker *α*-SMA in the lungs of C57BL/6 mice exposed to CuO NPs by intranasal delivery, was detected [20] suggesting the possible induction of pulmonary fibrosis. Similarly, we also found the upregulation of α-SMA expression. Our results therefore indicate that even a short, 3-day inhalation of CuO NPs, may negatively affect lung tissues. Longer inhalation periods were commonly characterized by changes of immune response-related biological processes and pathways. This observation is in agreement with previous studies [1,6,8], indicating the induction of local and systemic inflammatory responses after chronic CuO NPs exposures. We further focused on the identification of biological processes and pathways unique for 2 weeks, 6 weeks, and 3 months of inhalation, respectively. A two-week inhalation period affected the processes associated with apoptosis and programmed cell death. This result is comparable with the observations of Lai et al., who found time- and dose-dependence increased the intensity of the TUNEL-positive cells in the lung sections of mice, after the intranasal delivery of CuO NPs [20]. A six-week exposure period impacted on cell cycle-related processes. Although no similar results in mice have been reported, it has been shown that CuO NP may cause a cell cycle arrest in human keratinocytes, mouse embryonic fibroblasts [21], and in human umbilical vein endothelial cells (HUVECs) [22]. In a model of poorly differentiated hepatocellular carcinoma cells, the cell cycle arrest was possibly caused by histone H2AX phosphorylation, resulting from DNA damage [17]. The longest 3-month inhalation period caused changes in cell adhesion, cell migration, and related processes. These processes depend on extracellular matrix organization and their misregulation is implicated in tumorigenesis [23]. Although no relevant in vivo animal studies have been published, in vitro experiments in model cell lines suggest the inhibition of cell migration after exposure to CuO NPs [21,22].

To the best of our knowledge, our study is the first one to investigate miRNA expression changes after CuO NPs inhalation in mice. We found relatively low numbers of differentially expressed miRNAs that increased with the length of inhalation. For the short (3 day) exposure period no significant results were observed. Although no reports on miRNA expression after CuO NPs inhalation have been published, some of the miRNAs we detected were implicated after exposure to other NPs. Specifically, the expression of mmu-miR-21a-5p, mmu-miR-147-3p, mmu-miR-449a-5p, and mmu-miR-703 was deregulated in the lungs of mice exposed to multi-walled carbon nanotubes [24] and mmu-miR-342-3p in mice exposed to nano-TiO2 particles [25]. The deregulation of the miRNAs observed in our study was further found in the lungs of mice after cigarette smoke exposure (mmu-miR-21a-5p, mmu-miR-135b-5p, mmu-miR-146a-5p, mmu-miR-146b-5p, mmu-miR-155-5p, mmu-miR-342-3p, mmu-miR-449a-5p, mmu-miR-489-3p, and mmu-miR-672-5p) [26], in mice with cigarette smoke induced atherosclerosis (mmu-miR-21a-5p and mmu-miR-155-5p) [27], in the lungs of mice after radon inhalation (mmu-miR-21a-3p, mmu-miR-135b-5p, mmu-miR-146b-5p, mmu-miR-1298-5p, mmu-miR-6972-5p, and mmu-miR-703) [28], in mice with lung tumors (mmu-miR-135b-5p, mmu-miR-146b-5p, mmu-miR-223-3p, and mmu-miR-1298-5p) [29,30,31], in an ovalbumin-induced asthma mouse model (mmu-miR-21a-5p, mmu-miR-135b-5p, mmu-miR-146a-5p, mmu-miR-155-5p, mmu-miR-223-3p, and mmu-miR-451a) [32], and in mice with pneumonia and influenza infections (mmu-miR-21a-5p, mmu-miR-144-5p, mmu-miR-155-5p, and mmu-miR-223-3p) [33,34,35]. This overall suggests that the inhalation of CuO NPs induces a similar response as the above-mentioned adverse conditions and therefore likely has negative effects on the organism.

To further study the biological impacts of CuO NPs, we searched for miRNA–mRNA interactions by correlating the miRNA expression with mRNAs significantly deregulated by the inhalation. In contrast to the number of deregulated mRNAs that was lower after 6 weeks than 2 weeks of exposure, the number of interactions increased with the exposure time. In addition, the number of miRNAs targeting mRNAs was highest after 6 weeks of inhalation. Although it is difficult to explain these discrepancies, we may speculate that involvement of piRNAs detected in these interactions after 6 weeks of inhalation plays a role. piRNAs have been shown to be expressed in a tissue-specific manner and modulate various signaling pathways on both transcriptional and post-transcriptional manner [36]. Unlike miRNAs, they preferentially target transposons. Their abnormal expression in various cancers has been reported [36].

The results of miRNA–mRNA interactions analysis became a basis for the assessment of deregulated KEGG pathways. These pathways, specific for the response to CuO NPs exposure, partly overlapped with the deregulated biological processes and pathways identified for mRNA expression. Such pathways included e.g., ECM receptor interaction and focal adhesion (3-day exposure) or immune response related pathways (3-month inhalation). In addition, some unique pathways have been identified. Deregulation of Axon guidance (2 weeks) suggests changes in cell migration, which are potentially linked to cancer development [37]. Changes in Gap junction pathway (2 weeks) may affect intercellular communications which are frequently dysregulated in cancer [38]. Cell–cell communication deregulation is further related to Wnt signaling pathway (2 weeks) [39]. Calcium signaling, a key mechanism for the intercellular communication that is altered in lung cancer [40] also shows potential adverse effects of CuO NPs inhalation. The association of CuO NPs inhalation with cancer is further supported by the fact that pathways in cancer were deregulated after 3 days and 2 weeks of exposure. The deregulation of dilated cardiomyopathy pathway is most likely linked to changes in the immune response in the exposed mice [41]. The induction of olfactory transduction pathway after 2 weeks and 3 months of treatment is unexpected, considering the fact that non-olfactory tissues were investigated. However, it has been shown that mouse pulmonary macrophages can express olfactory receptors as a result of their activation by interferon-***γ*** [42]. Thus, the deregulation of this pathway may again reflect changes in immune response following the inhalation. Finally, we should mention the deregulation of lysosome pathway, detected after 6 weeks and 3 months of exposure. This pathway is related to synthesis of lysosomal enzymes that help to degrade foreign structures in the cells. In a recent study, SiO_2_ NPs were found in the lysosomes of macrophages of exposed mice [43]. We can therefore expect that CuO NPs inhalation affected the lysosome pathway in our study.

We performed global DNA methylation analysis to identify the possible effects of CuO NPs inhalation on the epigenome but found no differences between the exposed and control animals in any of the time intervals. Global DNA methylation has recently been investigated by Lu et al. in mice exposed to CuO NPs by intratracheal instillation [44]. In contrast to our results, the authors found increased levels of 5-hydroxymethylcytosine in the lungs after 24-h exposure. In another study, CuO NPs had no effect on DNA methylation of selected inflammation-related genes [10]. Thus, CuO NPs have the potential to modify DNA methylation levels, but the response depends on the mode of administration and length of exposure. It should also be noted that the results may depend on the method used to analyze DNA methylation: a gene-specific approach would most likely yield a different outcome than the global DNA methylation analysis.

## 5. Conclusions

CuO NPs are used in many industrial products and as antimicrobial agents. Although their production is lower than other metal-based NPs, their toxicity is high. They have been shown to induce oxidative stress, apoptosis, and cell cycle arrest in vitro. In laboratory animals exposed to CuO, NPs inflammation was most commonly detected, but neurotoxicity was also noted. Our study focused on transcriptomic and epigenetic changes in mice following CuO NPs inhalation and revealed distinct effects depending on the length of exposure. We observed a deregulation of extracellular matrix organization, impacts on immune response, cell cycle regulation, cell adhesion and apoptosis, as well as pathways potentially implicated in carcinogenesis. Due to the potential negative health effects of exposure in humans, more studies are needed to better characterize the mechanisms of CuO NPs toxicity in more detail.

## Figures and Tables

**Figure 1 nanomaterials-10-00550-f001:**
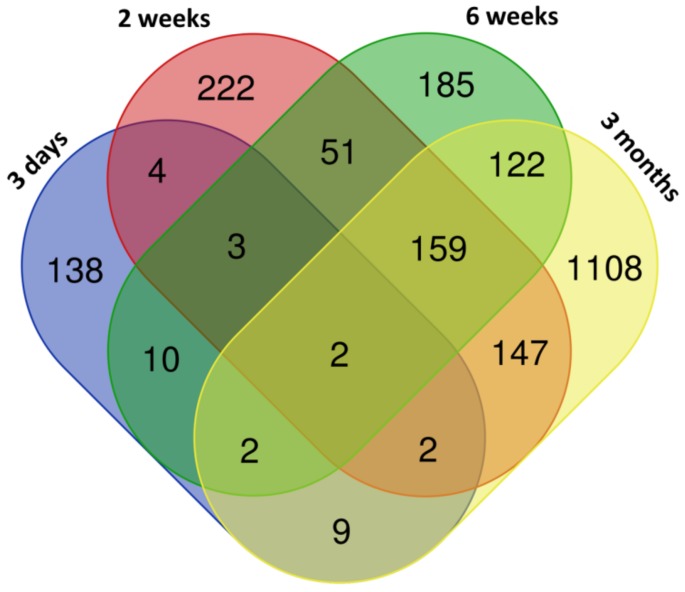
The distribution of common and unique genes differentially expressed after copper(II) oxide nanoparticles (CuO NPs) inhalation.

**Figure 2 nanomaterials-10-00550-f002:**
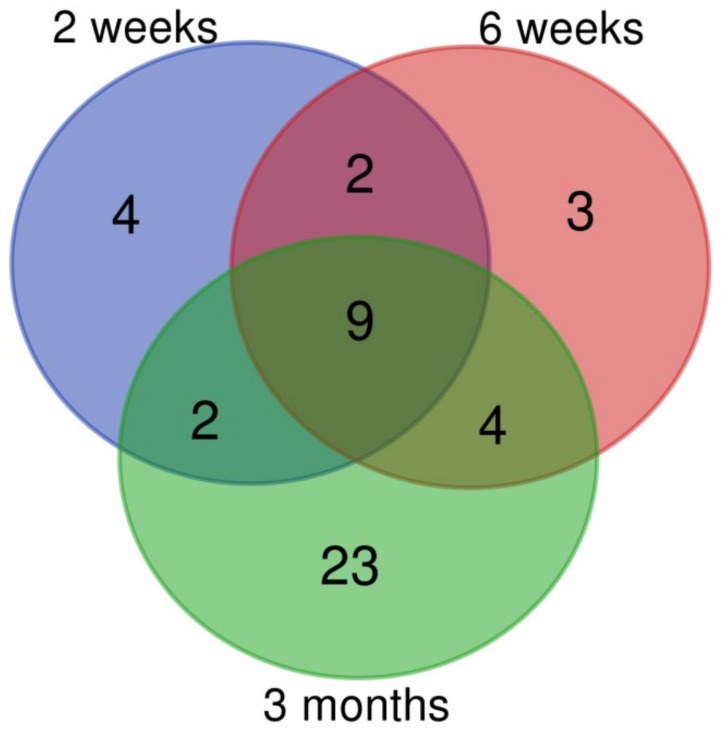
The numbers of common and unique miRNAs deregulated after CuO NPs inhalation.

**Figure 3 nanomaterials-10-00550-f003:**
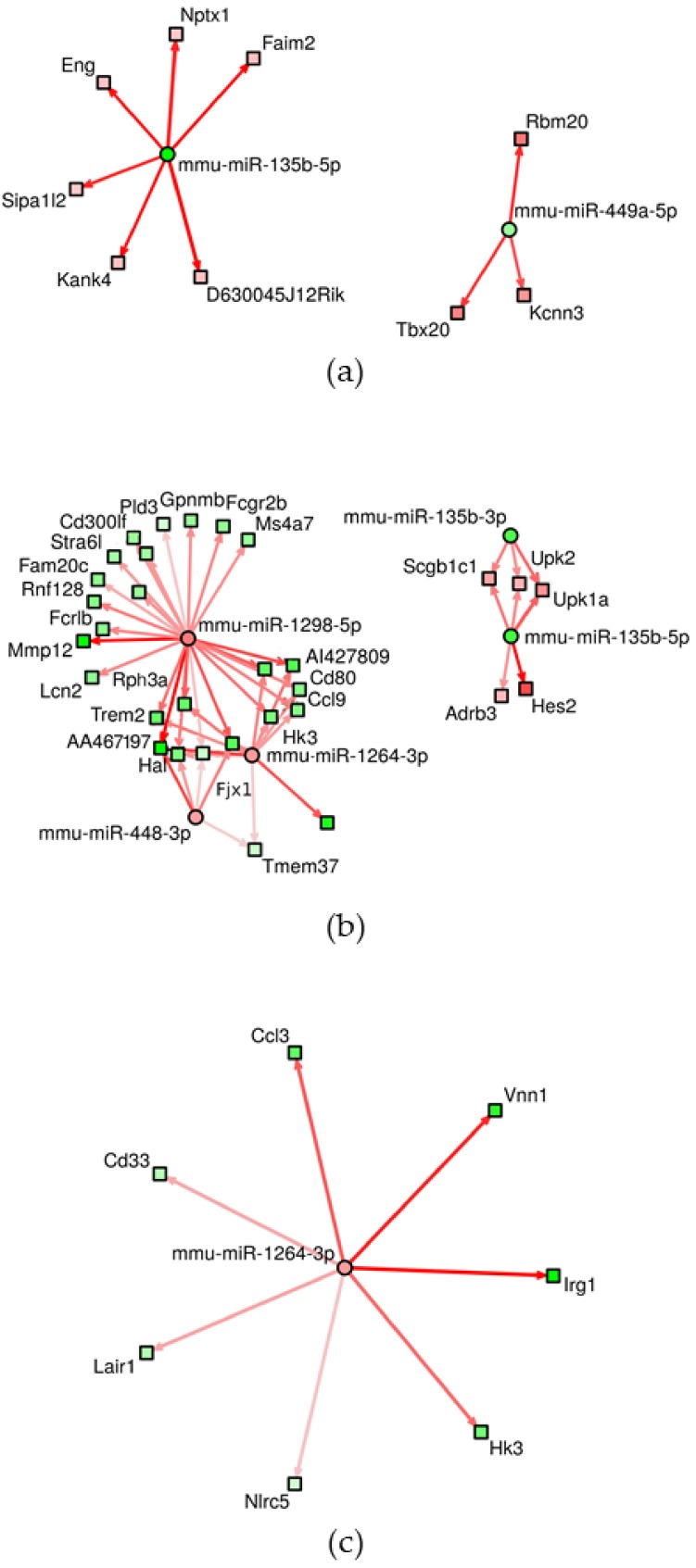
Network of miRNA–mRNA interactions after 2 weeks (**a**), 6 weeks (**b**), and 3 months (**c**) of inhalation. The results for individual miRNAs and their targets for adjusted *p*-value <0.001 and score ≥3 are visualized. Circles represent miRNAs, squares mRNAs. Red fill indicates upregulation, green fill corresponds to downregulation. The lines depict miRNA–mRNA pairs [13].

**Figure 4 nanomaterials-10-00550-f004:**
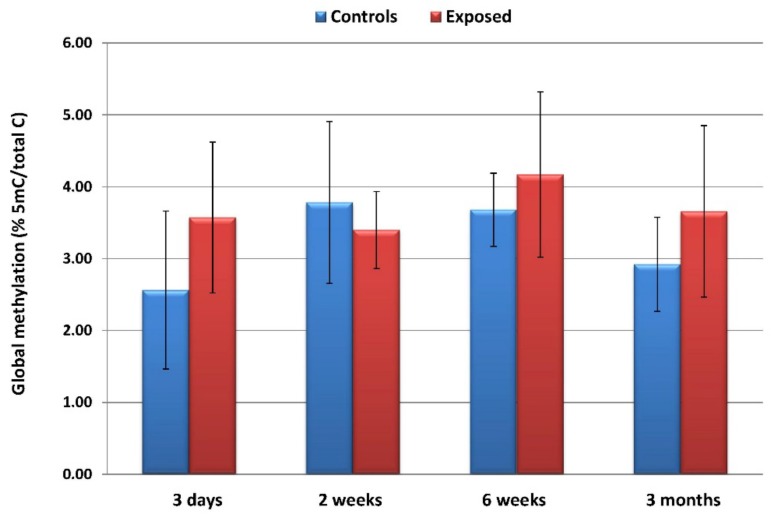
**The average (±SD)** global DNA methylation levels reported as a proportion of 5-methylcytosine/total cytosine in the exposed and control samples.

**Table 1 nanomaterials-10-00550-t001:** Selected biological processes and pathways deregulated after inhalation of CuO NPs.

ID	Deregulated Biological Process/Pathway	*q*-Value	Deregulated Genes (*N*)	Genes in Pathway (*N*)
**3 Days**				
GO:0030574	collagen catabolic process	0.003	7	67
GO:0043062	extracellular structure organization	0.005	13	355
GO:0030198	extracellular matrix organization	0.005	13	354
GO:0032963	collagen metabolic process	0.028	7	122
GO:0061077	chaperone-mediated protein folding	0.028	5	52
1269985	Chondroitin sulfate biosynthesis	0.009	4	22
82982	Glycosaminoglycan biosynthesis - chondroitin sulfate/dermatan sulfate	0.009	4	20
545276	chondroitin sulfate biosynthesis (late stages)	0.009	3	8
1270245	Collagen formation	0.019	6	93
1270247	Assembly of collagen fibrils and other multimeric structures	0.019	5	60
**2 Weeks**				
GO:0006955	immune response	< 0.001	165	1572
GO:0002682	regulation of immune system process	< 0.001	140	1506
GO:0006952	defense response	< 0.001	142	1651
GO:0002684	positive regulation of immune system process	< 0.001	102	976
GO:0001775	cell activation	< 0.001	96	1001
1269203	Innate Immune System	< 0.001	101	1312
1470924	Interleukin-10 signaling	< 0.001	21	49
83051	Cytokine-cytokine receptor interaction	< 0.001	41	270
1457780	Neutrophil degranulation	< 0.001	50	492
1269310	Cytokine Signaling in Immune system	< 0.001	64	763
**6 Weeks**				
GO:0006955	immune response	< 0.001	122	1572
GO:0002682	regulation of immune system process	< 0.001	113	1506
GO:0006954	inflammatory response	< 0.001	76	711
GO:0006952	defense response	< 0.001	115	1651
GO:0001816	cytokine production	< 0.001	70	700
1269203	Innate Immune System	< 0.001	97	1312
1457780	Neutrophil degranulation	< 0.001	56	492
1470924	Interleukin-10 signaling	< 0.001	17	49
M5889	Ensemble of genes encoding extracellular matrix and extracellular matrix-associated proteins	< 0.001	70	1028
M5885	Ensemble of genes encoding ECM-associated proteins including ECM-affiliated proteins, ECM regulators and secreted factors	< 0.001	57	753
**3 Months**				
GO:0006955	immune response	< 0.001	282	1572
GO:0006952	defense response	< 0.001	277	1651
GO:0051240	positive regulation of multicellular organismal process	< 0.001	260	1610
GO:0002682	regulation of immune system process	< 0.001	249	1506
GO:0001816	cytokine production	< 0.001	154	700
1269203	Innate Immune System	< 0.001	210	1312
1457780	Neutrophil degranulation	< 0.001	105	492
1269310	Cytokine Signaling in Immune system	< 0.001	125	763
1470924	Interleukin-10 signaling	< 0.001	25	49
83051	Cytokine-cytokine receptor interaction	< 0.001	62	270

**Table 2 nanomaterials-10-00550-t002:** Selected unique biological processes deregulated after inhalation of CuO NPs.

ID	Deregulated Biological Process	*q*-Value	Deregulated Genes (*N*)	Genes in Pathway (*N*)
**2 Weeks**				
GO:0006955	immune response	0.019	33	1572
GO:0045776	negative regulation of blood pressure	0.019	6	56
GO:0010941	regulation of cell death	0.019	33	1650
GO:0002682	regulation of immune system process	0.019	31	1506
GO:0043067	regulation of programmed cell death	0.022	31	1536
GO:0002683	negative regulation of immune system process	0.022	14	420
GO:0012501	programmed cell death	0.022	36	1952
GO:0001775	cell activation	0.025	23	1001
GO:0042981	regulation of apoptotic process	0.025	30	1519
GO:0006915	apoptotic process	0.027	35	1923
GO:0006955	immune response	0.019	33	1572
**6 Weeks**				
GO:0000280	nuclear division	< 0.001	24	599
GO:0048285	organelle fission	< 0.001	24	636
GO:1903047	mitotic cell cycle process	< 0.001	26	931
GO:0000278	mitotic cell cycle	< 0.001	27	1016
GO:0022402	cell cycle process	< 0.001	32	1385
GO:0000070	mitotic sister chromatid segregation	< 0.001	10	136
GO:0051241	negative regulation of multicellular organismal process	0.001	27	1127
GO:0051301	cell division	0.001	20	668
GO:0007049	cell cycle	0.001	35	1766
GO:0055118	negative regulation of cardiac muscle contraction	0.001	3	4
**3 Months**				
GO:0051240	positive regulation of multicellular organismal process	< 0.001	171	1610
GO:0022610	biological adhesion	< 0.001	164	1542
GO:0007155	cell adhesion	< 0.001	161	1530
GO:0040011	locomotion	< 0.001	170	1735
GO:0016477	cell migration	< 0.001	137	1300
GO:0001816	cytokine production	< 0.001	90	700
GO:0051674	localization of cell	< 0.001	141	1428
GO:0048870	cell motility	< 0.001	141	1428
GO:0006928	movement of cell or subcellular component	< 0.001	170	1882
GO:0009611	response to wounding	< 0.001	106	967
GO:0051240	positive regulation of multicellular organismal process	< 0.001	171	1610

**Table 3 nanomaterials-10-00550-t003:** miRNAs and numbers of corresponding targeted mRNAs after inhalation of CuO NPs for individual exposure periods.

miRNA	Target mRNA (N)
**2 Weeks**	
mmu-miR-135b-5p	50
mmu-miR-147-3p	11
mmu-miR-449a-5p	87
**6 Weeks**	
mmu_piR_017289/gb/DQ696831/Mus_musculus:17:27580702:27580732:Minus	1
mmu_piR_017289/gb/DQ696831/Mus_musculus:18:60298547:60298577:Plus	1
mmu_piR_017289/gb/DQ696831/Mus_musculus:3:46956408:46956438:Minus	2
mmu_piR_017289/gb/DQ696831/Mus_musculus:6:128725729:128725759:Minus	4
mmu-miR-1264-3p	134
mmu-miR-1298-5p	183
mmu-miR-135b-3p	22
mmu-miR-135b-5p	68
mmu-miR-146b-5p	11
mmu-miR-147-3p	13
mmu-miR-155-5p	15
mmu-miR-21a-5p	5
mmu-miR-448-3p	104
mmu-miR-489-3p	1
mmu-miR-672-5p	2
mmu-miR-6972-5p	4
mmu-miR-703	1
mmu-miR-7048-5p	4
**3 Months**	
mmu-miR-1264-3p	515
mmu-miR-1298-5p	461
mmu-miR-135b-5p	29
mmu-miR-144-5p	249
mmu-miR-146a-5p	23
mmu-miR-146b-5p	15
mmu-miR-147-3p	26
mmu-miR-155-5p	34
mmu-miR-21a-3p	43
mmu-miR-21a-5p	11
mmu-miR-21b	11
mmu-miR-223-3p	8
mmu-miR-342-3p	12
mmu-miR-3967	14
mmu-miR-451a	91

**Table 4 nanomaterials-10-00550-t004:** Kyoto Encyclopedia of Genes and Genomes (KEGG) pathways based on miRNA–mRNA interactions deregulated after inhalation of CuO NPs.

ID	Deregulated Biological Process/Pathway	*q*-Value	Deregulated Genes (*N*)	Genes in Pathway (*N*)
**3 Days**				
mmu04510	Focal adhesion	<0.001	19	199
mmu04512	ECM receptor interaction	<0.001	11	88
mmu05200	Pathways in cancer	0.003	21	539
**2 Weeks**				
mmu04520	Adherens junction	<0.001	27	71
mmu04510	Focal adhesion	<0.001	44	199
mmu04270	Vascular smooth muscle contraction	<0.001	29	140
mmu04360	Axon guidance	<0.001	30	180
mmu04916	Melanogenesis	<0.001	24	100
mmu05414	Dilated cardiomyopathy	<0.001	23	94
mmu05200	Pathways in cancer	<0.001	51	539
mmu04330	Notch signaling pathway	<0.001	16	54
mmu04540	Gap junction	<0.001	21	86
mmu04810	Regulation of actin cytoskeleton	<0.001	36	216
mmu04740	Olfactory transduction	<0.001	5	1134
mmu04912	GnRH signaling pathway	<0.001	21	90
mmu04310	Wnt signaling pathway	<0.001	26	162
mmu05217	Basal cell carcinoma	<0.001	14	63
mmu04020	Calcium signaling pathway	0.001	27	192
mmu05410	Hypertrophic cardiomyopathy (HCM)	0.001	17	91
mmu04512	ECM receptor interaction	0.002	16	88
mmu05412	Arrhythmogenic right ventricular cardiomyopathy (ARVC)	0.002	15	77
mmu05213	Endometrial cancer	0.003	11	58
mmu04144	Endocytosis	0.003	27	269
mmu11190	Dorso-ventral axis formation	0.005	7	22
mmu05211	Renal cell carcinoma	0.005	13	68
mmu05016	Huntingtons disease	0.005	25	199
mmu04720	Long term potentiation	0.006	13	67
mmu04260	Cardiac muscle contraction	0.010	14	86
**6 Weeks**				
mmu04142	Lysosome	<0.001	23	124
**3 Months**				
mmu04142	Lysosome	<0.001	43	124
mmu04740	Olfactory transduction	<0.001	4	1134
mmu04060	Cytokine-cytokine receptor interaction	<0.001	60	296
mmu05140	Leishmania infection	<0.001	27	69
mmu04062	Chemokine signaling pathway	<0.001	45	198
mmu04650	Natural killer cell mediated cytotoxicity	<0.001	34	117
mmu04612	Antigen processing and presentation	<0.001	23	91
mmu05332	Graft versus host disease	<0.001	15	65
mmu04662	B cell receptor signaling pathway	<0.001	22	82
mmu04666	Fc gamma R-mediated phagocytosis	<0.001	26	90
mmu04620	Toll like receptor signaling pathway	<0.001	25	99
mmu05330	Allograft rejection	<0.001	14	63
mmu16848	Epithelial cell signaling in helicobacter pylori infection	<0.001	21	69
mmu04940	Type I diabetes mellitus	<0.001	15	70
mmu04664	Fc epsilon RI signaling pathway	0.001	20	67
mmu05320	Autoimmune thyroid disease	0.002	14	78
mmu05416	Viral myocarditis	0.006	17	88
mmu04660	T cell receptor signaling pathway	0.006	23	103
mmu04640	Hematopoietic cell lineage	0.008	18	95
mmu04672	Intestinal immune network for IgA production	0.009	12	42

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
