# Peer review of "Gene Expression and Epigenetic Changes in Mice Following Inhalation of Copper(II) Oxide Nanoparticles"

_nanomaterials, 2020, doi:10.3390/nano10030550_

Round 1
Reviewer 1 Report
In this article the authors present a study of biological effects induced by the inhalation of copper oxide nanoparticles on mice. They mimicked acute and chronic exposures and analysed gene expression and epigenetic changes. In this work the authors go further in the investigation with respect to their previous work or study already present in the literature (e.g. Costa et al.). The results are well presented, I would recommend the article for publication.
Author Response
In this article the authors present a study of biological effects induced by the inhalation of copper oxide nanoparticles on mice. They mimicked acute and chronic exposures and analysed gene expression and epigenetic changes. In this work the authors go further in the investigation with respect to their previous work or study already present in the literature (e.g. Costa et al.). The results are well presented, I would recommend the article for publication.
Response: We thank the reviewer for this comment.
Reviewer 2 Report
In their article „Gene expression and epigenetic Changes in Mice Following Inhalation of Copper Oxide Nanoparticles, Rossner et al. exposed mice to conditions mimicking acute and chronic inhalation of CuO nanoparticles. The authors carried out mRNA, miRNA and DNA methylation analysis to investigate the effects of CuO exposure. All in all, I think this is a very nice study demonstrating the effect CuO nanoparticles have on overall gene expression and highlights the need for further studies in this field to understand the effects of nanoparticle inhalation on gene expression.
I would suggest publication of the manuscript subject to addressing of the following minor points:
- The authors give numbers for the numbers of differentially expressed genes – were these numbers averaged over all 8 mice in that group? This is not clear. If it was not averaged, there should be some errors/standard deviations given. It would be interesting to see if the number was indeed similar over all mice.
- It would be helpful to see some more discussion about the relationship between the number of differentially expressed genes (mRNA deregulation lower after 6 weeks compared to 2 weeks) and miRNA-mRNA interactions (highest at 6 weeks).
- Not requesting further experiments for this publication, but, similar to the study by Costa et al. (ref [10] in the manuscript) it would be interesting to see how the genetic effects change after a recovery period.
Author Response
In their article „Gene expression and epigenetic Changes in Mice Following Inhalation of Copper Oxide Nanoparticles, Rossner et al. exposed mice to conditions mimicking acute and chronic inhalation of CuO nanoparticles. The authors carried out mRNA, miRNA and DNA methylation analysis to investigate the effects of CuO exposure. All in all, I think this is a very nice study demonstrating the effect CuO nanoparticles have on overall gene expression and highlights the need for further studies in this field to understand the effects of nanoparticle inhalation on gene expression.
I would suggest publication of the manuscript subject to addressing of the following minor points:
The authors give numbers for the numbers of differentially expressed genes – were these numbers averaged over all 8 mice in that group? This is not clear. If it was not averaged, there should be some errors/standard deviations given. It would be interesting to see if the number was indeed similar over all mice.
Response: The numbers of differentially expressed genes (DEGs) were reached with all the samples falling into the treatment groups under consideration. In particular, to obtain the DEG list for 3 days exposure, the expression for all the mice exposed for 3 days is compared with the expression observed for all the mice from the 3 days control group. In the revision, we provide standard deviations for the numbers of differentially expressed mRNAs as recommended by the reviewer. We estimated them with bootstrapping. We employed 25 times repeated sampling with replacement carried out independently in each of the treatment groups. The results suggest that standard deviations for these numbers are relatively large as a consequence of small sample sizes, however, the hypothesis about increasing number of DEGs with increasing exposure remains reasonable. We changed our manuscript in Sections 2.6 and 3.1 accordingly.
It would be helpful to see some more discussion about the relationship between the number of differentially expressed genes (mRNA deregulation lower after 6 weeks compared to 2 weeks) and miRNA-mRNA interactions (highest at 6 weeks).
Response: We checked these contradictory results and concluded that deregulation of piRNAs expression involved in the interactions after 6 weeks exposure might play a role. We modified Section 2.4, Table 3 and Discussion (Section 4) and updated the list of references.
Not requesting further experiments for this publication, but, similar to the study by Costa et al. (ref [10] in the manuscript) it would be interesting to see how the genetic effects change after a recovery period.
Response: We agree with the reviewer that such approach would bring interesting data. It may be used in future studies.